# Associations between Gross and Fine Motor Skills, Physical Activity, Executive Function, and Academic Achievement: Longitudinal Findings from the UK Millennium Cohort Study

**DOI:** 10.3390/brainsci14020121

**Published:** 2024-01-24

**Authors:** Yuxi Zhou, Andrew Tolmie

**Affiliations:** Motor-Executive Control Interaction Lab, Department of Psychology and Human Development, UCL Institute of Education, University College London, London WC1E 6BT, UK; yuxi.zhou.21@ucl.ac.uk

**Keywords:** executive function, motor skills, physical activity, academic achievement, Millennium Cohort Study

## Abstract

Accumulating evidence from behavioral studies and neuroscience suggests that motor and cognitive development are intrinsically intertwined. To explore the underlying mechanisms of this motor–cognition link, our study examined the longitudinal relationship of early motor skills and physical activity with later cognitive skills. The sample was 3188 children from the United Kingdom Millennium Cohort Study, followed at 9 months and 5, 7, and 11 years. Early motor skills were examined at 9 months. Children’s daily physical activity level was measured using accelerometers at 7 years and a questionnaire was conducted at 11 years. Cognitive skills, including executive function and academic achievement, were measured at age 11. The results suggest that gross motor skills were positively associated with spatial working memory, whereas fine motor skills were predictive of good English and science outcomes. Moderate-to-vigorous activity was found to be negatively associated with English performance, although self-reported activity frequency was positively linked to math. Our results highlight the significant role of both gross and fine motor skills in cognitive development. This study also elucidates the limitations of using activity intensity to assess the impact of motor activity on children’s cognitive development, suggesting that attention to the effects of specific types of physical activity would better elucidate the motor/cognition link.

## 1. Introduction

Executive function (EF) refers to a family of higher-order cognitive processes that enable goal-directed behaviors, such as problem solving, planning, and reasoning [1], which have been consistently reported to be strongly associated with academic achievement [2]. In addition, EF is also identified as a critical predictor of various life outcomes such as wealth and physical/psychological health to an even greater extent than socioeconomic status or IQ [3,4]. According to Miyake et al. [5], there are three core EFs, inhibition, working memory/updating, and cognitive flexibility/shifting, all of which are considered to underlie cognitive processes that facilitate learning [6]. Given its significance, substantial resources have been invested in designing/implementing interventions to promote EF growth in children. However, the transfer of skills from such training has almost uniformly been found to be limited (e.g., computerized inhibitory control games [7]), and researchers have suggested that to exert an optimal effect, interventions targeting the full range of EF components and influences on these are needed, such as motor training [1]. Accumulating evidence from behavioral studies and neuroscience research has suggested that motor and cognitive development are intrinsically intertwined [8]. This is evident from studies investigating children with attention deficit–hyperactivity disorder (ADHD) and developmental coordination disorder (DCD), which have suggested that cognitive difficulties often coexist with motor deficits and vice versa [9]. The idea of motor–cognition associations is historically rooted in Piaget’s theory that the emergence of sensorimotor and motor abilities is a precursor for cognitive development [10]. In a similar vein, current embodied cognition theory suggests that sophisticated cognitive control did not evolve primarily for thinking or cognition but to better control actions in response to ever-changing demands for survival [11]. This viewpoint is further supported by evidence from developmental neuroscience research, indicating that brain regions responsible for motor functions (cerebellum) and EF (prefrontal cortex) exhibit similar developmental trajectories and that there is close co-activation between subregions of the cerebellum and prefrontal cortex during the performance of cognitive tasks [8,12]. Thus, these different lines of evidence provide compelling evidence for the close correspondence between motor and EF development. It is therefore reasonable to assume that higher levels of motor skills in early years confer an advantage for EF that may persist at least into adolescence.

Preliminary evidence of this specific link was established by the Northern Finland 1966 Birth Cohort Study, which documented significant predictive effects of earlier walking onset on EF 35 years later [13,14]. This predictive effect has also been found to extend to early old age, as evidenced in another more recent birth cohort study [15]. The results of MRI analysis in Ridler’s study [14] also suggest that frontal cortico–cerebellar systems associated with adult EF were anatomically linked to systems correlated with infant motor functions. This suggests that faster maturation of motor systems in the early years may have a positive impact on the development of the more complex brain circuits involved in higher-order cognitive processes later [13]. Embodied cognition perspectives provide a possible mechanism, indicating that better gross motor skills in the early years increase opportunities for motor engagement with the world, enriching the experience of the need to exert cognitive control [16].

Despite these links, the mechanisms underlying the relationships between early motor skills and cognitive skills are not yet well understood, and there is little clear mapping of the relationships themselves. Specifically, there is no conclusive evidence regarding which aspects of motor skills contribute to specific cognitive skills [17]. Gross motor (GM) skills refer to movement involving large muscle groups, whereas fine motor (FM) skills refer to movement involving small muscle groups. A series of cross-sectional studies conducted on children from diverse age groups and ethnic backgrounds have consistently reported that GM ability is positively linked to EF in children [18,19,20,21]. In comparison, research examining the specific roles of FM skills in EF development is scarce, and the available results are contradictory [22,23]. One longitudinal study examined early motor skills and EF in 3-year-old children by considering both GM and FM skills and their relations with more diverse EF measures [24]. The researchers found that the overall motor score assessed at time-point 1 (1 year old) has little effect on all aspects of EF. However, separate scores for GM and FM skills measured at time-point 2 (2 years old) exhibit different patterns of associations with different EF tasks, with better GM skills predicting increased inhibition and working memory and FM ability merely predicting working memory. Although the generalizability of this study is limited due to its small sample size and questionable control measures (early general cognitive ability), it offers insights into the important and potentially distinctive roles of both GM and FM abilities in later EF and emphasizes the need for the inclusion of more sophisticated measures for motor functions and EF.

As for academic achievement, the predictive role of FM skills in academic success has been empirically confirmed, especially with math and reading [25,26,27,28,29]. However, there is a heated debate as to whether GM skills also play a significant role [30]. Although the longitudinal studies mentioned earlier only reported significant results for FM skills and not GM skills [25,26,27], studies focusing on specific components of GM skills have revealed that certain aspects, such as balance and coordination, are significantly linked to mathematical skills [31,32,33,34]. Thus, this indicates that different GM skills may have a discrete influence on academic outcomes. Further insights into the possible GM function/academic achievement associations have been offered by recent studies [35,36], which suggest that GM ability may be indirectly linked to academic achievement through EF. This seems plausible given the close link between the cerebellum and the prefrontal cortex, although more research is needed to permit a solid conclusion to be drawn regarding this mediating effect. In conclusion, while existing empirical studies have found some associations between gross and fine motor abilities and cognitive skills, current findings are largely inconsistent, and the nature of the influence of the early years (cf. the embodied cognition account) is unclear. This is partly attributable to the fact that only a small number of studies have evaluated the impact of GM and FM skills simultaneously, and it is challenging to compare the results of different studies due to the varying measures used for both motor skills and cognitive skills.

In addition to fundamental motor skills, another aspect of motor function that has received extensive focus is physical activity (PA). Recent meta-analyses and systematic reviews have confirmed the general link between PA and cognitive abilities in children [37,38,39,40], but as in the motor skills/cognition research, consensus has not yet been established regarding which types of PA exert beneficial effects on which particular cognitive functions [41]. For example, Egger et al. [42] conducted a classroom-based intervention study with 142 7- to 9-year-old children. For 20 weeks, children were assigned to three training groups with varying levels of PA intensity and cognitive engagement: the *aerobics group* (high PA intensity and low cognitive demand), the *cognitive group* (low PA intensity and high cognitive demand), and the *combo group* (high PA exertion and high cognitive engagement). Performance in EF and academic outcomes were compared before and after the training. The results suggested that children in the aerobics group showed no improvement in either EF or general academic performance, whereas those in the group with high PA and high cognitive demand (combo group) showed significant improvement in both EF and math abilities. This finding supports the view that only cognitively enriched physical exercises enhance children’s EF [43,44]. However, opponents argue that moderate-intensity exercises are sufficient to induce a positive impact on EF [37,38], and the classroom-based PA in Egger et al.’s study was limited in intensity and duration and therefore not sufficient to generate discernable effects on children’s EF. Furthermore, Egger et al.’s study also found that training with high levels of PA was beneficial for mathematical skills regardless of the involvement of cognitive engagement. This positive link between PA and academic outcomes has been frequently reported in previous studies [45,46,47,48]. However, since these studies employ various intervention strategies, such as classroom-based PA programs, aerobics dance, acute physical exercise, and moderate-to-vigorous activity (MVPA), it is difficult to compare results and draw conclusions about whether and which kinds of PA are beneficial for academic outcomes. Additionally, most of the existing studies in physical activity training and cognition have been criticized as poor quality, with issues including insufficient sample sizes, failure to include control groups, and problematic measurement tools [41]. More systematic research with bigger samples is thus needed to further elucidate the associations between PA and cognition, enabling more evidence-based intervention programs to be designed and implemented for promoting children’s overall cognitive functions.

At the same time, previous randomized controlled studies have pointed out that pure aerobic exercises (e.g., MVPA) can significantly improve EF and academic outcomes in primary-school children [49,50,51]. The favorable effect of MVPA on academic success was also observed in Mullender-Wijnsma’s study, although the researchers found that the relationship between MVPA and academic achievement followed opposite directions in younger and older children, with second-grade children scoring lower in math following the intervention and third-grade children demonstrating better math outcomes. The author concluded that the MVPA/academic achievement relationship might be age-dependent [48]. Whilst these studies have demonstrated the positive effect of MVPA training on cognitive ability in preadolescent children utilizing the “gold-standard” RCT method, the generalizability of these results is still limited as it remains questionable whether cognitive gains endure after training has been completed. More importantly, very few researchers have examined the effect of habitual everyday MVPA on children’s cognitive development; examining the effects of naturally occurring physical activities may provide valuable insights into the mechanisms underlying the PA/cognitive associations over time and also how pre-existing motor abilities relate to PA [52].

Taken together, then, previous studies have highlighted possible inter-relationships between motor skills, physical activity, cognitive abilities, and academic outcomes; however, specific relationships among these concepts remain unclear. Few studies have examined the respective roles of GM and FM abilities in children’s EF and academic achievement, and the variety of measures employed in different studies makes it challenging to draw definitive conclusions on how GM and FM skills are related to different aspects of cognition. Also, the lack of longitudinal studies precludes inferring causal relationships from the available research. Furthermore, while MVPA training may have beneficial effects on children’s cognitive abilities, whether its beneficial effect extends to MVPA in natural settings is unknown. Hence, this study sought to fill the gap in the literature by utilizing data from a large-scale birth cohort study, namely the Millennium Cohort Study (MCS). The MCS collected prospective data on cohort members’ gross and fine motor skills in infancy, physical activity at ages 7 and 11 years (including naturally occurring MVPA), and also EF and academic achievement as outcome variables at 11 years of age. Our aim was to examine the effect of infant GM and FM skills and physical activity on subsequent cognitive skills and academic outcomes, and also the mediating role of EF among these relationships. We hypothesized that (1) both GM and FM skills would predict EF and academic performance; (2) physical activity would also have a beneficial effect on EF and academic outcomes; and (3) EF would mediate the effects of motor skills and PA on academic outcomes.

## 2. Materials and Methods

### 2.1. Participants and Analytic Sample

Data were drawn from the UK’s Millennium Cohort study, which tracks around 19,000 children born in 2000/2002 throughout their lives. The initial data were collected from families with children aged 9 months, with a total of 18,818 cohort members from 18,552 families (sweep 1). Eligible families were identified from the government record of Child Benefit. There have been 8 sweeps of data collection to date, which were conducted when cohort members were around 3, 5, 7, 11, 14, 17, and 22 years old (sweeps 2–8). The MCS employed a stratified random sampling method, which was designed to overrepresent children of ethnic minority backgrounds who resided in disadvantaged areas and children who grew up in smaller nations in the UK [53]. Original ethical approval was granted by the NHS Research Ethics Committee, and informed consent was obtained from parents and the cohort member themselves later. Most of the information was collected through interviews with the main respondent (overwhelmingly the mother), as well as self-completed questionnaires that were administered in the child’s home from age 7 onwards. The present study utilized data from sweeps when children were 9 months and 5, 7, and 11 years old (i.e., the first, third, fourth, and fifth sweeps, respectively). The current study was conducted in accordance with the Declaration of Helsinki, and the protocol was approved by the UCL IOE research ethics committee on 22 February 2022 (project identification code ZhouFeb2022). The data were accessed for research purposes after ethics approval had been granted and appropriately coded before being released to the researchers so that they were totally anonymous.

The analytic sample for the present study consisted of 3188 children with complete data on all variables analyzed (i.e., EF, academic achievement, early motor skills, PA, and three control variables) to ensure that the results were not misleading because of the inclusion of different sample sizes. At the first interview, participants were between approximately 8.6 and 12.7 months of age (*M*_age_ = 9.80, *SD* = 13.75), and 48.2% were male. The majority of participants in our study were of White ethnicity (87.6%), while the ethnic breakdown of the remaining participants was as follows: Indian (2.5%), Pakistani and Bangladeshi (3.4%), Black or Black British (2.1%), other ethnic groups (including Chinese; 1.3%), and mixed ethnicity (2.9%).

### 2.2. Measures

There were three sets of predictor variables: gross and fine motor skills, accelerometry-based PA, and self-reported PA, measured at sweeps 1, 4, and 5, respectively. The outcomes of interest, EF and academic achievement, were both measured at the fifth sweep. Covariates included family income and maternal education (measured at the first sweep) and verbal and nonverbal ability (measured at the third sweep).

#### 2.2.1. Motor Skills at 9 Months

Children’s gross and fine motor skills at 9 months were reported by the main respondent (usually the mother). Eight motor milestones of the Denver Developmental Screening Test were used to measure the gross and fine motor skills in infants [54]. Four items focused on FM skills, including grabbing objects, passing a toy, picking up small objects, and putting hands together. The remaining four items assessed GM skills, including sitting without support, moving from one place to another, standing up alone, and walking a few steps (see Appendix A). The main respondents were required to rate the frequency at which the child exhibited the corresponding behavior (1 = *not yet*, 2 = *once or twice*, 3 = *often*). The items for FM and GM skills were added up separately to yield the final score for the two abilities, with higher scores representing a higher level of development.

#### 2.2.2. Physical Activity at 7 and 11 Years

Habitual daily MVPA was measured between 2008 and 2009 using Actigraph GT1M accelerometers (Actigraph, Pensacola, FL, USA), when the cohort members were seven years old. They were instructed to wear the accelerometers on their right hip during waking hours for seven consecutive days, except during water-based activities. Participants who wore the accelerometer for at least 10 h on at least two days were considered to have valid data. Accelerometers were programmed to record activity at 15 s intervals, reporting the amplitude and frequency of acceleration events within given epoch as “counts” (a device-specific arbitrary unit) [55]. Time in moderate-to-vigorous physical activity (MVPA) was defined as greater than 2241 accelerometer counts per minute for 7-year-old children [56]. Since each child wore the device for a varying number of valid days, the mean time spent in MVPA was used for the analysis in this study. At 11 years of age, self-reported PA was assessed by children reporting the frequency with which they engaged in physical exercises (i.e., “how often do you play sports or active games inside or outside?”), from *never* (=1) to *most days* (=5).

#### 2.2.3. Executive Function at 11 Years

The computer-based Cambridge Neuropsychological Test Automated Battery (CANTAB [57]) was used to assess children’s spatial working memory (SWM) and decision making at age 11. The CANTAB allows for the measurement of processing speed in cognitive testing, which permits a greater sensitivity to children’s processing capability than accuracy alone [58]. It is comparable to traditional neuropsychological testing instruments, and consistent studies have shown that it is appropriate for assessing EF in children [59,60,61].

Decision making. Decision making was measured with the Cambridge Gambling Task (CGT [57]). Participants were presented with a row of ten boxes at the top of the screen (colored red and/or blue), one of which contained a yellow token. The task’s aim was to accurately guess where the yellow token is hidden in five stages, each consisting of several blocks of trials. In the decision making stage, the children were required to decide where the token was hidden and register their answer by touching the response box labeled “red” or “blue” at the bottom of the screen. In the second stage (gambling stage), participants were given 100 points and were asked to select a proportion of these points to gamble on their confidence in their decision. The possible bet values were displayed sequentially (either incrementally decreases or increases) in the center of the screen. Participants were informed that these points would be taken away or added to their total score and that their task was to win more points. The task generated six outcomes, but only “quality of decision making” was included in the analysis as the others were not significantly correlated with the main variables. Quality of decision making is the mean proportion of trials where the participant selects the correct color outcome, with higher scores representing better performance. The ability to make rational decisions under emotion salient context is closely associated with everyday motor behaviors, as is evidenced by studies suggesting that it is significantly predicted by motor inhibition [62]. Poor affective decision making was also found to be linked to poor self-control and impaired ability to anticipate action outcomes [63].

Spatial working memory. The selection of the spatial aspect of working memory as the outcome measure was based on its strong correlation with motor learning in comparison to the verbal aspect [64]. The Cambridge SWM Task was used to assess spatial working memory capacity [57]. During the task, participants were presented with several colored boxes on the screen, some of which contained blue tokens. The task was to find the blue tokens by touching each box and then moving the blue tokens to fill up the empty column on the screen. The participant needed to continue searching for tokens in the other boxes that had not previously contained a token until all tokens had been found in the simultaneously presented boxes. Participants were instructed to avoid revisiting boxes previously known to be empty or that already contained blue tokens, as these would be marked as error responses. The task difficulty increased gradually, from four to a maximum of eight boxes to search, with varying colors and locations of the boxes in each trial to avoid stereotyped searching strategies. Total errors and mean response time were the outcome measures used in the study, with higher scores indicating poorer SWM performance. Total errors refer to the frequency of all error types, and mean response time is the average time from the onset of each trial to the touch response on the final blue token. 

#### 2.2.4. Academic Achievement at 11 Years

The academic achievement of the cohort members was evaluated by teachers when they were 11 years old, and performance in four subjects (i.e., English, mathematics, science, and physical education) was included in the analysis [65]. Teachers rated children’s performance on a five-point scale from *well below average* (=1) to *well above average* (=5), with a higher score indicating better academic outcome.

#### 2.2.5. Control Variables

Control variables that correlated with both EF and academic performance were included as covariates in subsequent analyses to rule out alternative processes contributing to the outcome of interest [66]. These included socioeconomic status (SES) and early verbal and nonverbal ability [67,68,69].

Socioeconomic status. Family income and maternal education were included as two indicators for SES, which were reported by parents when children were nine months old. Family income was adjusted for the composition of the household using the Organization for Economic Cooperation and Development (OECD) equivalence scale. This scale generated new scores for each family, with higher values representing more equivalent income [65]. Maternal education was collected by asking the mothers to report their highest educational level (e.g., O level/GCSE grades A-C, first degree, or higher degree), which was coded as a continuous variable from *none of these qualifications* (=1) to *higher degree* (=8).

Verbal and nonverbal ability. Verbal and nonverbal abilities were measured using three subscales from the British Ability Scale II [70] when children were five years old: naming vocabulary (verbal), picture similarity (nonverbal), and pattern construction (nonverbal) subtests. Raw scores were used for the current analysis, and higher scores represent higher levels of ability.

#### 2.2.6. Analytic Strategy

A series of hierarchical regression analyses were conducted to examine the predictive effect of earlier motor coordination and PA on subsequent EF and academic success, as well as the mediating role of EF in the effects of motor skills and PA. Three models were fitted for EF (CGT decision making quality, SWM errors, and SWM response time) and four for academic achievement (English, math, science, and PE) (see Table 1 and Table 2). Predictors and covariates were entered in the order of measurement time to test for longitudinal effects on the two outcomes. Multivariate outliers were checked using the Mahalanobis distance (α > 0.05 [71]). As the model fit indices and coefficient parameters were quite similar with and without multivariate outliers, only analyses that include those outliers are reported to avoid improper inference from the results [72]. The significance of the results was determined at the 0.05 alpha level, and the R-square value was used to make inference on the model fit, with an R^2^ < 0.13, 0.13 ≤ R^2^ < 0.26, and R^2^ > 0.26 being considered a weak, moderate, and substantial effect, respectively [73].

## 3. Results

### 3.1. Preliminary Analyses

Table 3 presents descriptive statistics for all variables analyzed, and Table 4 displays the correlation coefficients among these variables. GM ability was significantly correlated with SWM reaction time (r = −0.06, *p* < 0.001) and PE performance (r = 0.12, *p* < 0.001) but not with other aspects of EF and academic outcomes. FM ability was not associated with any measures of EF, but it was positively linked to performance in all subjects (r ranged between 0.05 and 0.09). As for activity level, MVPA had a positive correlation with SWM total errors (r = 0.04, *p* = 0.043) and a negative correlation with the quality of decision making (r = −0.06, *p* = 0.002), but no significant association with SWM reaction time was detected. Negative associations were found between MVPA and performance in three core subjects (r ranged between −0.12 and −0.06), except that the link between MVPA and PE was positive (r = 0.18, *p* < 0.001). No significant relationship was found between self-reported PA and any measures of EF, but it was positively correlated with math outcome (r = 0.04, *p* = 0.02) and PE (r = 0.26, *p* < 0.001). All EF measures and academic outcomes were significantly related to each other, with better EF being associated with superior academic achievement (r ranged between 0.08 and 0.40).

### 3.2. Main Analyses

Prior to interpreting the results, several assumptions were tested. Firstly, an inspection of the scatterplot of standardized residuals against standardized predicted values indicated that the assumption of the normality, linearity, and homoscedasticity of residuals was met. High tolerance for all predictors indicated that multicollinearity would not interfere with the interpretation of the outcome.

#### 3.2.1. Gross and Fine Motor Skills, Physical Activity, and EF

For our study, we built three separate regression models to investigate the impact of infant motor skills and physical activity on later EF at age 11 (decision making quality, SWM reaction time and total errors). The model fit indices are shown in Table 5. SES and verbal and nonverbal abilities were entered as covariates in steps 1 and 4, respectively. As suggested in Table 6, GM skills significantly predicted SWM reaction time (∆R^2^ = 0.004, *p* < 0.001), and the predicting effect remained significant even after including the verbal and nonverbal abilities in the model (95%CI = [−440.740, −72.172]). It is noteworthy that pattern construction was a significant predictor of SWM reaction time, which appeared to account for unique variations beyond the influence of GM function, suggesting that GM ability is distinct from pattern retention and analysis. However, we failed to detect the significant predicting effects of FM skills on any of the EF outcomes. Furthermore, although adding MVPA at step 5 led to a significant R-square change in the model with decision making quality as the dependent variable, the predictive effect of MVPA was not significant, as the confidence interval contained zero (95%CI = [−0.001, 0.000]). As with FM skills, self-reported PA had little impact on any EF measures. Hence, our findings suggest that early GM ability is the sole significant predictor of EF, and it only predicts part of the EF measure (i.e., SWM reaction time). Regression results for SWM total errors and decision making quality could be found in the Appendix A.

#### 3.2.2. Gross and Fine Motor Skills, Physical Activity, EF, and Academic Achievement

Four separate regression analyses were conducted to examine the effect of infant motor skills, physical activity, and EF on academic outcomes in English, math, science, and PE at age 11. Table 7 presents the model fit indices for all regression models. The order of variables entered into the model for each academic subject was the same as that used for EF measures, although the EF measures were all entered in the last step. The results in Table 8 indicate that GM ability was a significant predictor of PE performance (∆R^2^ = 0.01, *p* < 0.001) but had little impact on performance in other subjects (95%CI = [0.03, 0.08]). In comparison, the results reveal positive effects of FM skills on both English and science performance (∆R^2^ = 0.006, *p* < 0.001 and ∆R^2^ = 0.003, *p* = 0.002 for English and science, respectively) but not math and PE outcomes (English: 95%CI = [0.04, 0.11]; science: 95%CI = [0.01, 0.07]; see Table 9 and Table 10).

Furthermore, a significant impact of MVPA on English and PE performance was detected (English: ∆R^2^ = 0.007, *p* < 0.001; PE: ∆R^2^ = 0.034, *p* < 0.001). However, while MVPA was positively predictive of PE performance (95%CI = [0.00, 0.01]), the relationship between MVPA and English was found to be negative (95%CI = [−0.01, −0.00]; Table 9). In step 6, the self-reported PA was entered in the model and found to have a significant predictive effect on both math and PE performance (math: ∆R^2^ = 0.001, *p* = 0.004; PE: ∆R^2^ = 0.049, *p* < 0.001). In contrast to MVPA, self-reported PA was a positive predictor of both (math: 95%CI = [0.00, 0.07]; PE: 95%CI = [0.17, 0.23]; see Table 11).

All EF measures were entered in the final step, showing a significant predicting effect on performance in all subjects (English: ∆R^2^ = 0.042, *p* < 0.001; math: ∆R^2^ = 0.081, *p* < 0.001; science: ∆R^2^ = 0.050, *p* < 0.001; PE: ∆R^2^ = 0.015, *p* < 0.001). The quality of decision making was found to predict all four academic outcomes (English: 95%CI = [0.288, 0.645]; math: 95%CI = [0.44, 0.78]; science: 95%CI = [0.23, 0.53]; PE: 95%CI = [0.04, 0.35]). The same predictive effect was found for SWM total error (English: 95%CI = [−0.01, −0.02]; math: 95%CI = [−0.02, −0.01]; science: 95%CI = [−0.01, −0.01]; PE: 95%CI = [−0.01, −0.00]), while SWM response time was linked to all subjects except for English performance (math: 95%CI = [−0.00, −0.00]); science (95%CI = [−0.00, −0.00]); PE (95%CI = [−0.00, −0.00]). No mediating effects of EF were detected, as evidenced by the lack of change in the beta values for any of the motor measures when EF was included. This indicates that these motor and EF skills are essentially independent contributors to academic outcomes, despite the relationship between GM skills and SWM reaction time.

## 4. Discussion

This study aimed to investigate the longitudinal relationships between gross and fine motor abilities, physical activity, and subsequent EF and academic achievement, using data from a large-scale birth cohort study. The main findings reveal that GM and FM skills were significantly linked to spatial working memory and academic achievement, respectively. However, the relationship between the time spent in MVPA and academic outcomes was found to be negative. Prior to the interpretation of the results, it is important to note that the effect sizes for all significant outcomes were small, which could possibly be attributed to the relatively insensitive measures (e.g., information on self-reported PA was collected using a single question) and the extended period of data collection, which may have introduced excessive noise into the results. Despite the modest R^2^ values, however, the low *p*-values suggest that the identified effects are sufficiently reliable to provide a deeper understanding of the phenomena investigated [74].

### 4.1. How Are Infant Motor Skills Linked to Later Cognitive Functions?

The first goal of this study was to examine the associations between infant motor skills and later cognitive abilities at age 11, while considering the effects of both GM and FM skills. Our results suggest that only infant GM skills, not FM skills, were related to later EF, specifically SWM reaction time. The significant GM/EF relationships found in this study are consistent with prior research that suggested a closer link between GM ability and working memory, particularly visuospatial working memory, and that this association becomes more robust when processing speed is taken into account [18,19,21,22,24,75,76]. This lends support to our expectation that earlier maturation of the motor system confers a favorable and lasting effect on later EF. Additionally, our results extend previous research focusing solely on the walking/standing onset [13,14], providing new insights into the long-term effect of overall gross motor control on children’s EF development. Notably, the specific link between GM skills and SWM found in the current study appears to have a distinct influence on pattern construction, which might stem from the nature of the SWM task and the motor control it demands. More specifically, the computerized SWM tasks used in this study involved dragging squares and placing them on the appointed position on the screen, which requires superior locomotor skills for faster and more accurate performance [77]. Nevertheless, the overlap between motor and EF performance is consistent with the theory that EF emerged to support control of action within physical tasks [78]. These findings have important implications for interventions aimed at promoting EF, as they suggest that targeting early motor development, particularly gross motor control, may have positive effects on EF for prepubertal children.

However, the current study did not detect significant relationships between early FM ability and EF, which contradicts the previous study by Wu et al. [24], who found positive relationships between early FM skills (measured at 1 year old) and better performance in working memory span test at 3 years of age. Similar results have also been reported in previous cross-sectional studies focusing on preschool children and using the same measures for FM skills (dexterity subscale of the Movement Assessment Battery for Children-2), which reported significant relationships between the total FM score and global EF performance [23,79]. A probable explanation for the non-significant results in our study is that the FM/EF association is also task-specific and was not observed here due to the limited measures for FM ability and EF. For example, a more recent study by Maurer and Roebers [80] investigated the effect of both GM and FM skills on concurrent EF performance and suggested that only difficult gross motor tasks are related to EF, but both “easy” and difficult fine motor tasks are significantly associated with EF, although “easy” fine motor tasks are still difficult for preschool children. This finding is consistent with a previous neuroimaging study indicating that motor skill learning is coupled with increased activity of the dorsolateral prefrontal cortex—a key region responsible for EF—and that activation decreases as the behavior becomes automated and then increases when the reestablishment of cognitive control over automated behavior is required [81]. In other words, EF appears to only be involved in the process of learning new/difficult motor tasks or in motor adaptation, where attention and cognitive control are needed. This points to a possible mechanism underlying the relationships between motor and EF development, which is through automatization [82]. While this assumption needs to be validated in further studies, the results of this study highlight the need to use more refined measures other than global GM and FM skills in future research.

At the same time, the importance of FM skills for later academic outcomes was once again confirmed in the present study, particularly in English and science. Therefore, it is possible that the positive link between FM ability and EF reported in prior studies may simply reflect the close link between EF and academic success, and there are some overlaps between FM ability and EF, both contributing to academic performance. Although further research is needed to support this view, it is in line with the embodied account of EF development that there is no inherent difference between motor and cognitive functions [11]. Regarding the inconsistent relations between FM skills and English, science, and math, a possible explanation could be that FM ability is a strong predictor of literacy competencies (including writing), which are thought to be more correlated with English and science but less with mathematical skills [28]. However, contradictory findings suggest that some FM components (i.e., fine motor integration) are significantly related to mathematics competencies [27,83], which could partly be attributed to the common/shared neural pathways that coordinate visuospatial processes for both fine motor integration and mathematical skills [83]. In this account, the lack of association between FM and math skills in the present study may be due to the fact that the important period for FM development and influence may in some respects be later than that for GM skills, with FM skills in infancy not yet involving many spatial motor abilities [84].

In terms of the association between GM skills and academic outcomes, it was unexpected that GM ability was only associated with PE performance—which makes inherent sense—but not with other subjects. This is in fact in accordance with the results of other studies [25,26,27,85]. However, in the literature, contradictory findings argue that specific components of GM ability, such as balancing and bilateral coordination, are positively associated with reading and math abilities [31,32,33,34]. It is well established that GM ability is theoretically linked to academic performance, particularly in early childhood, as it enables children to sit upright, coordinate their hand movements while writing, and develop spatial skills through exploration, which is closely related to math performance [6]. Therefore, it is again emphasized that investigating specific components within GM skills, rather than the overall GM score, could contribute to a better understanding of how gross motor control is related to performance in different subjects.

### 4.2. Is Physical Activity Beneficial for Cognitive Development?

The second aim was to investigate the impact of physical activity, particularly MVPA, on children’s cognitive abilities. It was unexpected that we failed to detect any significant relationship between PA and EF. Our results suggest that neither MVPA at age 7 nor self-reported PA at age 11 were associated with EF, which contradicts previous RCT studies investigating similar age groups [49,50,51]. However, the beneficial effects observed in previous intervention studies may have been due to other factors beyond the effect of activity intensity, such as the involvement of different motor skills or the adaptation required in the PA intervention. This is supported by findings from more recent research, which suggest that only PA with appropriate levels of cognitive demand, not merely pure aerobic exercise, can promote EF growth in children [42,44,86,87]. One explanation for this beneficial effect is the physiological arousal caused by increased physical fitness, whereas evidence from developmental neuroscience studies suggests that the evident improvement in EF is more likely a result of the strong parallelism in the developmental trajectories of motor and cognitive functions [88]. In a similar vein, recent studies indicate that football and cognitively enriched exergame training can effectively improve children’s EF [89,90]. The common characteristic underlying these types of physical activity is that they require constant adaptation in response to task rules/demands. For example, in order to have better performance in football, children need to focus on the game, analyze the situation to formulate a better action plan, and also inhibit their current actions; all of these require flexible control of the movement, which has been theorized to be a significant part of EF. Therefore, the non-significant findings in our study highlight that daily habitual MVPA is apparently insufficient for enhancing children’s EF, and future research in this field should be undertaken to explore the effect of specific types of PA and types of motor skills involved in those activities so as to better understand the PA/EF relations.

In terms of academic outcomes, it is not unexpected that MVPA was inversely related to English performance, although self-reported PA was positively related to math. Regarding the negative MVPA/English relationship, similar findings were reported in a previous study, which suggested that the MVPA/academic achievement association may be negative in younger children and positive in older children [48]. A possible explanation for this is that among younger children, PA is commonly encouraged to a greater extent for those who are seen as less intellectually able, whereas for older children, access to PA is often a function of school type, with higher SES/more able students going to schools where PA is a more organized part of the curriculum. Findings on self-reported PA are consistent with previous cross-sectional studies [91]. It could be that the self-reported frequency of exercise level might reflect greater levels of math-related spatial skills (e.g., spatial thinking [92]) relative to gross measures of activity intensity (MVPA). However, further studies examining different aspects of motor skills and different types of PA are needed to verify this hypothesis.

### 4.3. Meditating Role of EF

Finally, the mediating role of EF in the effects of motor abilities and PA was not determined, as the beta values for GM and FM skills and PA barely changed when EF measures were added to the regression models for academic outcomes. In contrast, previous studies found that EF fully mediated the relationship between GM skills and academic achievement [35,36]. Failure to replicate previous findings may be due to the lack of significant evidence of a GM skills/academic achievement association in the current study, highlighting the need for more refined measurements to disentangle the interconnections between motor skills, EF, and academic success. Similar results have been documented regarding the non-significant mediating effects of EF on PA/academic achievement [91]. However, Visier-Alfonso argued that there were strong mediating effects of EF, but these effects may be more evident for certain EF domains (i.e., shifting and inhibition [93]). This could offer some explanation for the results in this study. Nevertheless, few studies have examined the mediating effects of EF, and therefore the extent to which PA affects academic achievement via EF remains to be investigated more systematically.

### 4.4. Strengths and Limitations

Our findings should be interpreted in light of several limitations, which are briefly listed. Firstly, while mothers’ prospective reports of infants’ motor abilities may be a reliable source of information, concurrent data from healthcare professionals may have been more accurate in capturing their motor skills. Secondly, our measure of EF was based solely on two tasks from the CANTAB, as these were the only ones available in the MCS, and the lack of measures of other EF aspects (such as inhibition and cognitive flexibility) may partly account for the insignificant results in our study. It should be noted that the CGT used is not a traditional measure of the cool EF commonly used in previous studies investigating motor–EF relations; it measures “hot” EF that concerns cognitive control under salient emotional context (e.g., emotion, motivation, and reward) [94,95]. However, Leshem et al. [96] pointed out that hot and cool EF are independent yet interrelated cognitive processes, and that impairment in one aspect may affect the development of the other. For example, a study focusing on children with developmental coordination disorder (DCD) suggests that deficits in motor learning/inhibition found in children with DCD may be the result of poorer inhibition control towards emotional cues. These children may consider novel or difficult motor tasks as aversive and therefore avoid such negative emotions by sticking with less physically intense activities, which may, in turn, decrease the opportunity for the development of cool EF benefit from coupled the motor–EF effect [97]. Thus, future studies should consider including both cool and hot EF while using more refined measures for motor skills to better understand the specific links between different motor activities and aspects of EF [96]. Finally, the data collected in sweep 4 of the MCS only recorded the intensity and duration of children’s physical activity (MVPA) and did not specify the type of activity in which they participated. The self-reported physical activity at age 11 was also measured by a single question about the frequency of physical activities, which precludes further investigation into the relationship between different types of physical activity and motor and cognitive skills. 

Despite these limitations, this study is, to our best knowledge, the first to examine the motor–EF associations by simultaneously considering infant motor ability, accelerometer-measured physical activity, EF, and academic achievement. Additionally, our study used a large-scale birth cohort study with a large sample size and allowed for the prospectively assessed influence of measures of variable outcomes. This enables more reliable inferences to be drawn from the data. Thus, this study contributes to the scarce literature by providing evidence on the longitudinal effects of both FM and GM ability and the effect of habitual everyday MVPA on later cognitive outcomes.

### 4.5. Clinical Application

The results of our study, along with the existing literature, provide compelling evidence that early motor development is associated with later cognitive abilities. Given this, it is critical that healthcare professionals identify infants with motor delays as early as possible and implement timely and effective interventions. This is essential to prevent potentially adverse effects on children’s cognitive development. In addition, the results of our findings support the argument that school curricula which emphasize the importance of motor skills in young children (e.g., Montessori education) contribute to the overall cognitive function of children [17]. As for physical activity, our results indicate that excessive exercise may not necessarily be beneficial for children’s executive function and academic success. Therefore, school administrators and parents should shift their focus away from the quantity of exercise and instead recognize the significance of the quality and type of physical activities [44].

## 5. Conclusions

In conclusion, our study highlights that infant GM and FM skills are differentially associated with EF and academic achievement. Specifically, GM ability was found to be more closely linked to executive function, whereas FM ability was linked to academic success. However, the underlying mechanisms of these relationships need to be further explored. An investigation of the specific roles of different components within fine and gross motor skills down to the level of actual task requirements might be more helpful in elucidating how early motor skills are related to cognitive development [17]. With regards to physical activity, we found that objectively measured MVPA duration was negatively linked to academic achievement. This finding indicates that gross measures of physical exercise are generally unhelpful, and further studies should shift focus from merely investigating the intensity of PA to examining the qualitative aspects of PA, such as types of activity, emotional and social components, and which types of motor control are involved (if any)—as with GM/FM ability—to clarify the PA/cognitive relationships [98].

## Figures and Tables

**Table 1 brainsci-14-00121-t001:** Regression models: executive function as DVs.

Models	Specification
Model 1	Family income and maternal education
Model 2	Model 1 + gross motor skills
Model 3	Model 2 + fine motor skills
Model 4	Model 3 + BAS naming vocabulary and picture similarity and pattern construction
Model 5	Model 4 + MVPA
Model 6	Model 5 + Self-reported PA

Note. Three EF measures (i.e., decision making quality, SWM reaction time, and SWM total errors) were entered as DVs in turn.

**Table 2 brainsci-14-00121-t002:** Regression models: academic achievement as DVs.

Models	Specification
Model 1	Family income and maternal education
Model 2	Model 1 + gross motor skills
Model 3	Model 2 + fine motor skills
Model 4	Model 3 + BAS naming vocabulary and picture similarity and pattern construction
Model 5	Model 4 + MVPA
Model 6	Model 5 + Self-reported PA
Model 7	Model 6 + Decision making quality and SWM reaction time and SWM total errors

Note. Academic achievement in four subjects (i.e., English, math, science, and PE) were entered as DVs in turn.

**Table 3 brainsci-14-00121-t003:** Descriptive statistics for study variables.

Variable	Mean	SD	95% Confidence Interval
Family income	356.66	208.91	[349.40, 363.91]
Maternal education	4.70	1.89	[4.63, 4.76]
Gross motor skills	9.57	1.13	[9.53, 9.61]
Fine motor skills	11.61	0.79	[11.58, 11.63]
Naming vocabulary	15.04	3.27	[14.92, 15.15]
Picture similarity	16.04	3.41	[15.93, 16.16]
Pattern construction	20.01	7.62	[19.75, 20.28]
MVPA (min)	62.56	22.20	[61.78, 63.33]
Self-reported PA	4.38	0.89	[4.35, 4.41]
Decision making quality	0.83	0.17	[0.82, 0.83]
SWM reaction time (ms)	28,523.70	6050.06	[28,313.60, 28,733.79]
SWM total errors	32.26	17.74	[31.64, 32.87]
English performance	3.55	0.96	[3.52, 3.58]
Math performance	3.61	0.96	[3.57, 3.64]
Science performance	3.55	0.82	[3.52, 3.58]
PE performance	3.39	0.78	[3.36, 3.41]
Valid N	3188		

**Table 4 brainsci-14-00121-t004:** Parametric correlation coefficients (r) for the relationship among motor skills, physical activity, EF, and academic achievement (N = 3188).

	1	2	3	4	5	6	7	8	9	10	11
1. Gross motor skills	--										
2. Fine motor skills	0.17 ***	--									
3. MVPA	0.10 ***	−0.02	--								
4. Self-reported PA	0.04 *	0.01	0.15 ***	--							
5. Decision making quality	−0.02	0.00	−0.06 **	−0.00	--						
6. SWM reaction time	−0.06 ***	−0.01	−0.01	−0.02	−0.11 ***	--					
7. SWM total errors	−0.01	−0.03	0.04 *	−0.00	−0.17 ***	0.42 ***	--				
8. English	0.01	0.09 ***	−0.12 ***	0.01	0.19 ***	−0.19 ***	−0.32 ***	--			
9. Math	0.03	0.05 **	−0.05 **	0.04 *	0.22 ***	−0.26 ***	−0.40 ***	0.75 **	--		
10. Science	0.02	0.07 ***	−0.06 **	0.03	0.19 ***	−0.22 ***	−0.34 ***	0.78 **	0.80 **	--	
11. PE	0.12 ***	0.05 **	0.18 ***	0.26 ***	0.08 ***	−0.13 ***	−0.17 ***	0.26 ***	0.31 ***	0.30 ***	--

Note. two-tailed; * *p* < 0.05; ** *p* < 0.01; *** *p* < 0.001.

**Table 5 brainsci-14-00121-t005:** Model summary: executive function as DVs.

					Change Statistics				
Model	R	R^2^	Adjusted R Square	Std. Error of the Estimate	R-Square Change	F Change	df1	df2	Sig. F Change
*DV: Decision making quality*									
Model 1	0.14	0.02	0.02	0.16	0.02	32.66	2	3185	<0.001 ***
Model 2	0.14	0.02	0.02	0.16	0.00	0.75	1	3184	0.388
Model 3	0.14	0.02	0.02	0.16	0.00	0.00	1	3183	0.984
Model 4	0.20	0.04	0.04	0.16	0.02	20.48	3	3180	<0.001 ***
Model 5	0.20	0.04	0.04	0.16	0.00	4.90	1	3179	0.027 *
Model 6	0.20	0.04	0.04	0.16	0.00	0.00	1	3178	0.993
F (9, 3178) = 14.86, *p* < 0.001									
*DV: SWM reaction time*									
Model 1	0.11	0.01	0.01	6015.16	0.01	19.54	2	3185	<0.001 ***
Model 2	0.13	0.02	0.02	6004.33	0.00	12.50	1	3184	<0.001 ***
Model 3	0.13	0.02	0.02	6005.23	0.00	0.05	1	3183	0.828
Model 4	0.25	0.06	0.06	5863.26	0.05	53.00	3	3180	<0.001 ***
Model 5	0.25	0.06	0.06	5863.47	0.00	0.77	1	3179	0.379
Model 6	0.25	0.06	0.06	5863.63	0.00	0.83	1	3178	0.363
F (9, 3178) = 23.88, *p* < 0.001									
*DV: SWM total errors*									
Model 1	0.22	0.05	0.05	17.30	0.05	82.09	2	3185	<0.001 ***
Model 2	0.22	0.05	0.05	17.30	0.00	0.80	1	3184	0.371
Model 3	0.22	0.05	0.05	17.30	0.00	0.92	1	3183	0.338
Model 4	0.34	0.11	0.11	16.72	0.06	76.82	3	3180	<0.001 ***
Model 5	0.34	0.11	0.11	16.72	0.00	0.63	1	3179	0.429
Model 6	0.34	0.11	0.11	16.72	0.00	0.06	1	3178	0.814
F (9, 3178) = 45.41, *p* < 0.001									

Note. * *p* < 0.05; *** *p* < 0.001. Model 1: family income and maternal education. Model 2: Model 1 + gross motor skills. Model 3: Model 2 + fine motor skills. Model 4: Model 3 + BAS naming vocabulary and picture similarity and pattern construction. Model 5: Model 4 + MVPA. Model 6: Model 5 + self-reported PA. *DV: Decision making quality* refers to the regression model with decision making quality as the dependent variable. *DV: SWM reaction time* refers to the regression model with SWM reaction time as the dependent variable. *DV: SWM total errors* refers to the regression model with SWM total errors as the dependent variable.

**Table 6 brainsci-14-00121-t006:** Regression estimates (β(SE)) of motor skills and physical activity with regard to SWM reaction time (significant predictors in bold).

	Model 1	Model 2	Model 3	Model 4	Model 5	Model 6
Family income	−0.08 (0.58) ***	−0.08 (0.58) ***	−0.08 (0.58) ***	−0.05 (0.58) **	−0.05 (0.58) **	−0.05 (0.58) *
Maternal education	−0.05 (64.70) *	−0.05 (64.62) *	−0.05 (64.71) *	−0.02 (64.55)	−0.02 (64.56)	−0.02 (64.57)
**Gross motor skills**		−0.06 (94.11) ***	−0.06 (95.51) ***	−0.05 (93.43) **	−0.05 (93.96) **	**−0.05 (93.99) ****
Fine motor skills			0.00 (136.52)	0.02 (133.46)	0.02 (133.52)	0.02 (133.53)
Naming vocabulary				−0.01 (36.16)	−0.01 (36.19)	−0.01 (36.20)
Picture similarity				−0.03 (33.90)	−0.04 (33.91)	−0.04 (33.91)
Pattern construction				−0.20 (15.15) ***	−0.20 (15.15) ***	−0.20 (15.15) ***
MVPA					−0.02 (4.72)	−0.01 (4.78)
Self-reported PA						−0.02 (121.70)
Constant	(292.15) **	(945.68) ***	(1702.90) ***	(1730.41)***	(1755.22)***	(1806.84) ***

Note. SE = Std. Error; * *p* < 0.05; ** *p* < 0.01; *** *p* < 0.001; β = standardized coefficients.

**Table 7 brainsci-14-00121-t007:** Model summary: academic achievement as DVs.

					Change Statistics				
Model	R	R^2^	Adjusted R Square	Std. Error of the Estimate	R-Square Change	F Change	df1	df2	Sig. F Change
*DV: English performance*									
Model 1	0.32	0.10	0.10	0.91	0.10	181.85	2	3185	<0.001 ***
Model 2	0.32	0.10	0.10	0.91	0.00	0.16	1	3184	0.686
Model 3	0.33	0.11	0.11	0.91	0.01	20.59	1	3183	<0.001 ***
Model 4	0.48	0.23	0.22	0.85	0.12	160.37	3	3180	<0.001 ***
Model 5	0.48	0.23	0.23	0.85	0.01	28.72	1	3179	<0.001 ***
Model 6	0.48	0.23	0.23	0.85	0.00	0.47	1	3178	0.495
Model 7	0.52	0.27	0.27	0.82	0.04	61.13	3	3175	<0.001 ***
F (12, 3175) = 100.07, *p* < 0.001									
*DV: math performance*									
Model 1	0.31	0.10	0.10	0.91	0.10	174.12	2	3185	<0.001 ***
Model 2	0.32	0.10	0.10	0.91	0.00	3.25	1	3184	0.072
Model 3	0.32	0.10	0.10	0.91	0.00	3.62	1	3183	0.057
Model 4	0.48	0.23	0.23	0.84	0.13	183.18	3	3180	<0.001 ***
Model 5	0.48	0.23	0.23	0.84	0.00	0.53	1	3179	0.466
Model 6	0.48	0.23	0.23	0.84	0.00	4.05	1	3178	0.044 *
Model 7	0.56	0.32	0.31	0.80	0.08	124.10	3	3175	<0.001 ***
F (12, 3175) = 121.70, *p* < 0.001									
*DV: science performance*									
Model 1	0.33	0.11	0.11	0.78	0.11	189.73	2	3185	<0.001 ***
Model 2	0.33	0.11	0.11	0.78	0.00	1.04	1	3184	0.308
Model 3	0.33	0.11	0.11	0.78	0.00	9.26	1	3183	0.002 **
Model 4	0.48	0.23	0.23	0.72	0.12	165.98	3	3180	<0.001 ***
Model 5	0.48	0.23	0.23	0.72	0.00	1.89	1	3179	0.169
Model 6	0.48	0.23	0.23	0.72	0.00	1.04	1	3178	0.308
Model 7	0.53	0.28	0.28	0.70	0.05	73.15	3	3175	<0.001 ***
F (12, 3175) = 103.85, *p* < 0.001									
*DV: PE performance*									
Model 1	0.19	0.04	0.04	0.76	0.04	58.35	2	3185	<0.001 ***
Model 2	0.22	0.05	0.05	0.76	0.01	47.44	1	3184	<0.001 ***
Model 3	0.22	0.05	0.05	0.76	0.00	1.95	1	3183	0.162
Model 4	0.24	0.06	0.06	0.75	0.01	10.45	3	3180	<0.001 ***
Model 5	0.35	0.09	0.09	0.74	0.03	119.18	1	3179	<0.001 ***
Model 6	0.38	0.14	0.14	0.72	0.05	182.67	1	3178	<0.001 ***
Model 7	0.40	0.19	0.16	0.71	0.02	19.39	3	3175	<0.001 ***
F (12, 3175) = 49.67, *p* < 0.001									

Note. * *p* < 0.05; ** *p* < 0.01; *** *p* < 0.001. Model 1: family income and maternal education; Model 2: Model 1 + gross motor skills; Model 3: Model 2 + fine motor skills; Model 4: Model 3 + BAS naming vocabulary and picture similarity and pattern construction; Model 5: Model 4 + MVPA; Model 6: Model 5 + self-reported PA; Model 7: Model 6 + decision making quality and SWM reaction time and SWM total errors. *DV: English performance* refers to the regression model with English outcome as the dependent variable. *DV: math performance* refers to the regression model with math outcome as the dependent variable. *DV: science performance* refers to the regression model with science outcome as the dependent variable. *DV: PE performance* refers to the regression model with P.E. outcome as the dependent variable.

**Table 8 brainsci-14-00121-t008:** Regression estimates (β(SE)) of motor skills, physical activity, and EF on PE performance (significant predictors in bold).

	Model 1	Model 2	Model 3	Model 4	Model 5	Model 6	Model 7
Family income	0.13 (0.00) ***	0.13 (0.00) ***	0.13 (0.00) ***	0.12 (0.00) ***	0.12 (0.00) ***	0.11 (0.00) ***	0.10 (0.00) ***
Maternal education	0.09 (0.01) ***	0.09 (0.01) ***	0.09 (0.01) ***	0.07 (0.01) ***	0.08 (0.01) ***	0.08 (0.01) ***	0.08 (0.01) ***
**Gross motor skills**		0.12 (0.01) ***	0.12 (0.01) ***	0.11 (0.01) ***	0.09 (0.01) ***	0.08 (0.01) ***	**0.08 (0.01) *****
Fine motor skills			0.03 (0.02)	0.02 (0.02)	0.03 (0.02)	0.02 (0.02)	0.03 (0.02)
Naming vocabulary				−0.01 (0.01)	0.00 (0.01)	−0.01 (0.00)	−0.01 (0.00)
Picture similarity				0.02 (0.00)	0.02 (0.00)	0.03 (0.00)	0.02 (0.00)
Pattern construction				0.09 (0.00) ***	0.10 (0.00) ***	0.10 (0.00) ***	0.06 (0.00) *
**MVPA**					0.19 (0.00) ***	0.15 (0.00) ***	**0.15 (0.00) *****
**Self-reported PA**						0.23 (0.02) ***	**0.23 (0.01) *****
**Decision making quality**							**0.04 (0.08) ***
**SWM reaction time**							**−0.04 (0.00) ***
**SWM total errors**							**−0.10 (0.00) *****
Constant	(0.04) **	(0.12) ***	(0.22) ***	(0.22) ***	(0.22) ***	(0.22) ***	(0.24) ***

Note. SE = Std. Error; * *p* < 0.05; ** *p* < 0.01; *** *p* < 0.001; β = standardized coefficients.

**Table 9 brainsci-14-00121-t009:** Regression estimates (β(SE)) of motor skills, physical activity, and EF on English performance (significant predictors in bold).

	Model 1	Model 2	Model 3	Model 4	Model 5	Model 6	Model 7
Family income	0.14 (0.00) ***	0.14 (0.00) ***	0.14 (0.00) ***	0.07 (0.00) ***	0.07 (0.00) ***	0.07 (0.00) ***	0.04 (0.00) *
Maternal education	0.23 (0.01) ***	0.23 (0.01) ***	0.22 (0.01) ***	0.15 (0.01) ***	0.15 (0.01) ***	0.15 (0.01) ***	0.13 (0.01) ***
Gross motor skills		0.01 (0.01)	−0.01 (0.01)	−0.02 (0.01)	−0.01 (0.01)	−0.01 (0.01)	−0.01 (0.01)
**Fine motor skills**			0.08 (0.02) ***	0.06 (0.02) ***	0.06 (0.02) ***	0.06 (0.02) ***	**0.06 (0.02) *****
Naming vocabulary				0.21 (0.01) ***	0.21 (0.01) ***	0.28 (0.01) ***	0.20 (0.01) ***
Picture similarity				0.09 (0.01) ***	0.09 (0.01) ***	0.09 (0.01) ***	0.07 (0.01) ***
Pattern construction				0.18 (0.00) ***	0.18 (0.00) ***	0.18 (0.00) ***	0.12 (0.00) ***
**MVPA**					−0.08 (0.00) ***	−0.09 (0.00) ***	**−0.08 (0.00) *****
Self-reported PA						0.01 (0.02)	0.01 (0.02)
**Decision making quality**							**0.08 (0.09) *****
SWM reaction time							−0.03 (0.00)
**SWM total errors**							**−0.18 (0.00) *****
Constant	(0.04) **	(0.14) ***	(0.26) ***	(0.25) *	(0.25) **	(0.26) **	(0.28) ***

Note. SE = Std. Error; * *p* < 0.05; ** *p* < 0.01; *** *p* < 0.001; β = standardized coefficients.

**Table 10 brainsci-14-00121-t010:** Regression estimates (β(SE)) of motor skills, physical activity, and EF on science performance (significant predictors in bold).

	Model 1	Model 2	Model 3	Model 4	Model 5	Model 6	Model 7
Family income	0.14 (0.00) ***	0.14 (0.00) ***	0.14 (0.00) ***	0.07 (0.00) ***	0.07 (0.00) **	0.07 (0.00) ***	0.04 (0.00) *
Maternal education	0.23 (0.01) ***	0.23 (0.01) ***	0.23 (0.01) ***	0.15 (0.01) ***	0.15 (0.01) **	0.15 (0.01) ***	0.13 (0.01) ***
Gross motor skills		0.02 (0.01)	0.01 (0.01)	−0.01 (0.01)	−0.01 (0.01)	−0.01 (0.01)	−0.01 (0.01)
**Fine motor skills**			0.05 (0.02) **	0.04 (0.02) *	0.04 (0.02) *	0.04 (0.02) *	**0.04 (0.02) ***
Naming vocabulary				0.21 (0.00) ***	0.20 (0.00) ***	0.20 (0.00) ***	0.20 (0.00) ***
Picture similarity				0.09 (0.00) ***	0.09 (0.00) ***	0.09 (0.00) ***	0.07 (0.00) ***
Pattern construction				0.19 (0.00) ***	0.19 (0.00) ***	0.19 (0.00) ***	0.13 (0.00) ***
MVPA					-0.02 (0.00)	-0.02 (0.00)	−0.02 (0.00)
Self-reported PA						0.02 (0.02)	0.02 (0.01)
**Decision making quality**							**0.08 (0.08) *****
**SWM reaction time**							**−0.06 (0.00) *****
**SWM total errors**							**−0.19 (.00) *****
Constant	(0.04) **	(0.12) ***	(0.22) ***	(0.21) ***	(0.22) ***	(0.22) ***	(0.24) ***

Note. SE = Std. Error; * *p* < 0.05; ** *p* < 0.01; *** *p* < 0.001; β = standardized coefficients.

**Table 11 brainsci-14-00121-t011:** Regression estimates (β(SE)) of motor skills, physical activity, and EF on math performance (significant predictors in bold).

	Model 1	Model 2	Model 3	Model 4	Model 5	Model 6	Model 7
Family income	0.16 (0.00) ***	0.16 (0.00) ***	0.16 (0.00) ***	0.10 (0.00) ***	0.10 (0.00) ***	0.10 (0.00) ***	0.06 (0.00) ***
Maternal education	0.20 (0.01) ***	0.20 (0.01) ***	0.20 (0.01) ***	0.13 (0.01) ***	0.13 (0.01) ***	0.13 (0.01) ***	0.10 (0.01) ***
Gross motor skills		0.03 (0.01)	0.03 (0.01)	0.01 (0.01)	0.01 (0.01)	0.01 (0.01)	0.01 (0.01)
Fine motor skills			0.03 (0.02)	0.02 (0.02)	0.02 (0.02)	0.02 (0.02)	0.02 (0.02)
Naming vocabulary				0.15 (0.01) ***	0.14 (0.01) ***	0.14 (0.01) ***	0.14 (0.01) ***
Picture similarity				0.09 (0.01) ***	0.09 (0.01) ***	0.09 (0.01) ***	0.07 (0.01) ***
Pattern construction				0.26 (0.00) ***	0.26 (0.00) ***	0.26 (0.00) ***	0.18 (0.00) ***
MVPA					−0.01 (0.00)	−0.02 (0.00)	−0.01 (0.00)
**Self-reported PA**						0.03 (0.02) *	**0.03 (0.02) ***
**Decision making quality**							**0.10 (0.09) *****
**SWM reaction time**							**−0.06 (0.00) *****
**SWM total errors**							**−0.24 (0.00) *****
Constant	(0.04) **	(0.14) ***	(0.26) ***	(0.25) ***	(0.25) ***	(0.26) ***	(0.27) ***

Note. SE = Std. Error; * *p* < 0.05; ** *p* < 0.01; *** *p* < 0.001; β = standardized coefficients.

## Data Availability

The datasets analyzed for this study can be found in the UK Data Service (https://beta.ukdataservice.ac.uk/datacatalogue/series/series?id=2000031, accessed on 1 March 2022). Data are available through the UK Data Service after approval by the CLS Data Access Committee.

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
