# Peer review of "Associations between Gross and Fine Motor Skills, Physical Activity, Executive Function, and Academic Achievement: Longitudinal Findings from the UK Millennium Cohort Study"

_brainsci, 2024, doi:10.3390/brainsci14020121_

Round 1
Reviewer 1 Report
Comments and Suggestions for Authors
My comments are:
· The authors should summarize the paper a little. The authors should be more specific in the presentation of the study in the paper
· The structure of the abstract is not correct. Authors must follow the journal's guidelines: objective, methods,...
· Abstract: There should be no limitations
· The discussion should begin with the main finding of the study.
· Authors must include a "Clinical Application" section within the discussion section.
· The conclusion must give a more concrete message. It must have an extension of 3 lines at most. It should well summarize the main finding of the study.
Comments on the Quality of English LanguageThe level of scientific English must be improved
Reviewer 2 Report
Comments and Suggestions for Authors
Having read the manuscript, I think the authors have done a good job. The methodology used is clearly explained, making it easy to understand how the study was carried out and the measurements and analyses performed. In addition, this study provides further evidence for the existing relationships between executive functions, physical activity, motor skills, and academic performance.
I have just one minor question. As stated in the study's limitations, why didn't the authors measure other executive functions, especially inhibition?
Reviewer 3 Report
Comments and Suggestions for Authors
This study aimed to explore potential associations between motor skills, physical activity, executive function, and academic achievement. The study is both original and relevant, making a significant contribution to Brain Sciences. However, I understand that the manuscript needs revision, and I present some suggestions and questions here for the proper application of findings.
1. Title: The title is adequate. However, I question whether including the target population would enhance its clarity for readers.
2. Abstract: The abstract section is well written and highlights the main findings of the study. I suggest the authors briefly explain the statistical procedures and include R-square values.
3. Keywords: Although not mandatory, I suggest the use of MESH terms.
4. Introduction: Avoid using extensive paragraphs, such as those seen in rows 30-59 and 111-142. Also, refrain from starting paragraphs with “In conclusion” (row 104).
5. Methods: I commend the authors for explaining each point of the methods used in this study, allowing a complete comprehension of the study protocol. Tables 1 and 2 are also important to clarify the statistical models.
6. Results: Include units for variables presented in Table 3. Instead of using minimum and maximum values, the 95% confidence interval would be more appropriate. Clarify whether you used Pearson or Spearman’s index in Table 4.
7. Discussion: When performing several regression models, could this increase the type-1 statistical error? Please discuss this topic.
Conclusion: Adequate.
References: Adequate.
Comments on the Quality of English LanguageMinor editing of English language required
